# Near infrared spectroscopic evaluation of biochemical and crimp properties of knee joint ligaments and patellar tendon

Jari Torniainen[1,2]*, Aapo Ristaniemi[1], Jaakko K. Sarin[1,2,3], Mithilesh Prakash[4], Isaac O. Afara[1,7], Mikko A. J. Finnilä[5], Lauri Stenroth[1], Rami K. Korhonen[1], Juha Töyräs[1,6,7]

**1** Department of Applied Physics, University of Eastern Finland, Kuopio, Finland, **2** Diagnostic Imaging Center, Kuopio University Hospital, Kuopio, Finland, **3** Department of Medical Physics, Medical Imaging Center, Pirkanmaa Hospital District, Tampere, Finland, **4** A. I. Virtanen Institute for Molecular Sciences, University of Eastern Finland, Kuopio, Finland, **5** Research Unit of Medical Imaging, Physics and Technology, Faculty of Medicine, University of Oulu, Oulu, Finland, **6** Science Service Center, Kuopio University Hospital, Kuopio, Finland, **7** School of Information Technology and Electrical Engineering, The University of Queensland, Brisbane, Australia

* jari.torniainen@uef.fi

**Data Availability Statement:** https://etsin.fairdata. fi/dataset/925999ad-8127-4428-b7cc-c03c12e62d3c https://doi.org/10.23729/40c4f149-

## Abstract

Knee ligaments and tendons play an important role in stabilizing and controlling the motions of the knee. Injuries to the ligaments can lead to abnormal mechanical loading of the other supporting tissues (e.g., cartilage and meniscus) and even osteoarthritis. While the condition of knee ligaments can be examined during arthroscopic repair procedures, the arthroscopic evaluation suffers from subjectivity and poor repeatability. Near infrared spectroscopy (NIRS) is capable of non-destructively quantifying the composition and structure of collagen-rich connective tissues, such as articular cartilage and meniscus. Despite the similarities, NIRS-based evaluation of ligament composition has not been previously attempted. In this study, ligaments and patellar tendon of ten bovine stifle joints were measured with NIRS, followed by chemical and histological reference analysis. The relationship between the reference properties of the tissue and NIR spectra was investigated using partial least squares regression. NIRS was found to be sensitive towards the water ($R2_{CV}$ = .65) and collagen ($R2_{CV}$ = .57) contents, while elastin, proteoglycans, and the internal crimp structure remained undetectable. As collagen largely determines the mechanical response of ligaments, we conclude that NIRS demonstrates potential for quantitative evaluation of knee ligaments.

## Introduction

Knee ligaments are fibrous connective tissues that provide mechanical stability to the knee by preventing excessive translation and rotation of the joint. The four main ligaments are anterior cruciate, posterior cruciate, lateral collateral, and medial collateral ligaments (i.e., ACL, PCL, LCL, and MCL, respectively). In addition, patellar tendon (PT), connecting patella to tibia, is often considered to be one of the knee ligaments as well. Ligaments consist primarily of water

d894-4126-bad2-79fe47b3e2fa Ristaniemi, A., Torniainen, J., & Paakkonen, T. (2021). Biomechanical, biochemical, and near infrared spectral data of bovine knee ligaments and patellar tendon (Version 1). Aapo Ristaniemi. https://doi.org/10.23729/40c4f149-d894-4126-bad2-79fe47b3e2fa.

**Funding:** Source: Valtion tutkimusrahoitus of Kuopio University Hospital (Kuopion Yliopistollinen Sairaala) Project: 5203111 URL: https://www.psshp.fi/tutkimus/tutkimusrahoitus/valtion-tutkimusrahoitus} Statement: The funders had no role in study design, data collection and analysis, decision to publish, or preparation of the manuscript. Recipient(s): Jari Torniainen Source: Academy of Finland Project: 315820 URL: https://www.aka.fi/en/ Statement: The funders had no role in study design, data collection and analysis, decision to publish, or preparation of the manuscript. Recipient(s): Isaac O. Afara Source: Sigrid Juselius Foundation (Sigrid Juseliuksen Säätiö) URL: https://www.sigridjuselius.fi/en/ Statement: The funders had no role in study design, data collection and analysis, decision to publish, or preparation of the manuscript. Recipient(s): Rami K. Korhonen, Aapo Ristaniemi Source: Academy of Finland Project: 324529 URL: https://www.aka.fi/en/ Statement: The funders had no role in study design, data collection and analysis, decision to publish, or preparation of the manuscript. Recipient(s): Rami K. Korhonen, Aapo Ristaniemi Source: Academy of Finland Project: 316258 URL: https://www.aka.fi/en/ Statement: The funders had no role in study design, data collection and analysis, decision to publish, or preparation of the manuscript. Recipient(s): Mithilesh Prakash.

**Competing interests:** The authors have declared that no competing interests exist.

(60–80% of total weight), collagen (75% of dry weight), elastin (10–15% of dry weight in ligaments, less than 3% of dry weight in tendons), proteoglycans (1–3% of dry weight), and fibroblasts (1.5% of dry weight) [1–3]. Main structural element of ligaments is collagen (primarily type I), which is hierarchically organized from collagen molecules into larger and larger interconnected fibrous bundles that eventually form the full ligament [2] (Fig 1). At rest, collagen fibers are folded into a crimp-like pattern which unfolds as the ligament is stretched [4]. In ligaments, the main role of collagen is to provide the tensile resistance against loading. Elastin modulates shear and transverse responses of ligaments and tendons [5, 6], stiffens the toe region response in uniaxial tension when collagen fibers are crimped [7, 8] and may increase Young's modulus at the linear region [8, 9]. Additionally, elastin is suggested to facilitate structural reorganization of the tissue after deformation, i.e. restore tissue back to its original shape [10]. Negatively charged proteoglycans attract water to the tissue [11, 12], affect collagen organization and sliding [13, 14], and influence the quasi-static and viscoelastic properties of ligaments [8, 15, 16]. Finally, fibroblasts are responsible for maintaining the extracellular matrix of the ligament [3, 17]. Each ligament has adapted to withstand loading along their longitudinal axis and the geometric arrangement of the four primary ligaments enables them to resist forces in all natural directions of loading.

Under extreme physical loading (e.g., sport accidents) ligaments, particularly the ACL, are at risk of suffering a partial or total mechanical failure. The resulting injury is often painful and can drastically change joint loading and homeostasis. The change in loading can induce more tensile, compressive, and shear stresses to other structures of the knee. Initial trauma and prolonged abnormal loading can lead to degeneration of other tissues like meniscus and articular cartilage and, in extreme cases, lead to osteoarthritis [18, 19]. Conservative treatment (physiotherapy) can be utilized in the case of ruptured ligaments, but often transplants (e.g., autografts from patellar tendon) are used to remedy the situation [20].

While the condition of ligaments can be non-invasively determined through magnetic resonance imaging [21], any repair procedure of ligaments must be performed arthroscopically. The minimally invasive orthopedic keyhole surgery is performed by an orthopedic surgeon by inserting an endoscopic camera and various surgical instruments into the joint cavity. In practice, the orthopedic surgeon will evaluate the damaged region, the extent of said damage and, in the case of an autograft, select the appropriate replacement tissue. For ACL grafts, closely matching the biomechanical and structural properties of the donor site to the injured ACL is of paramount importance [22]. During these procedures, the condition of various tissues at and around the injury are routinely evaluated by visual inspection (endoscopic camera) and by manual probing of the tissue (arthroscopic hook). Both of these methods are highly subjective, suffer from poor reproducibility, and could be greatly enhanced by a new, more quantitative method for assessing tissue integrity.

Near infrared spectroscopy (NIRS) is a non-destructive optical technique for performing bulk chemical analysis of different materials by measuring the vibrational absorbance of NIR light. A target sample is illuminated using a broadband light source and the reflected and backscattered light are collected by the spectrometer. The presence of different molecular bonds in the sample results in light absorption at different wavelengths. In other words, the NIR absorption spectrum is a representation of the chemical contents of the sample. Unlike mid infrared spectroscopy, the NIR spectrum is not intuitively interpretable visually but requires multivariate calibration models (e.g., partial least squares regression (PLSR) or principal component regression (PCR)) to relate the spectra to the investigated reference property. In contrast to other spectroscopic techniques, NIRS requires less sample preparation, provides deeper sample penetration, and is robust enough to be used in the field. NIRS has several industrial applications [23], such as chemical process control, pharmacology, and agriculture.

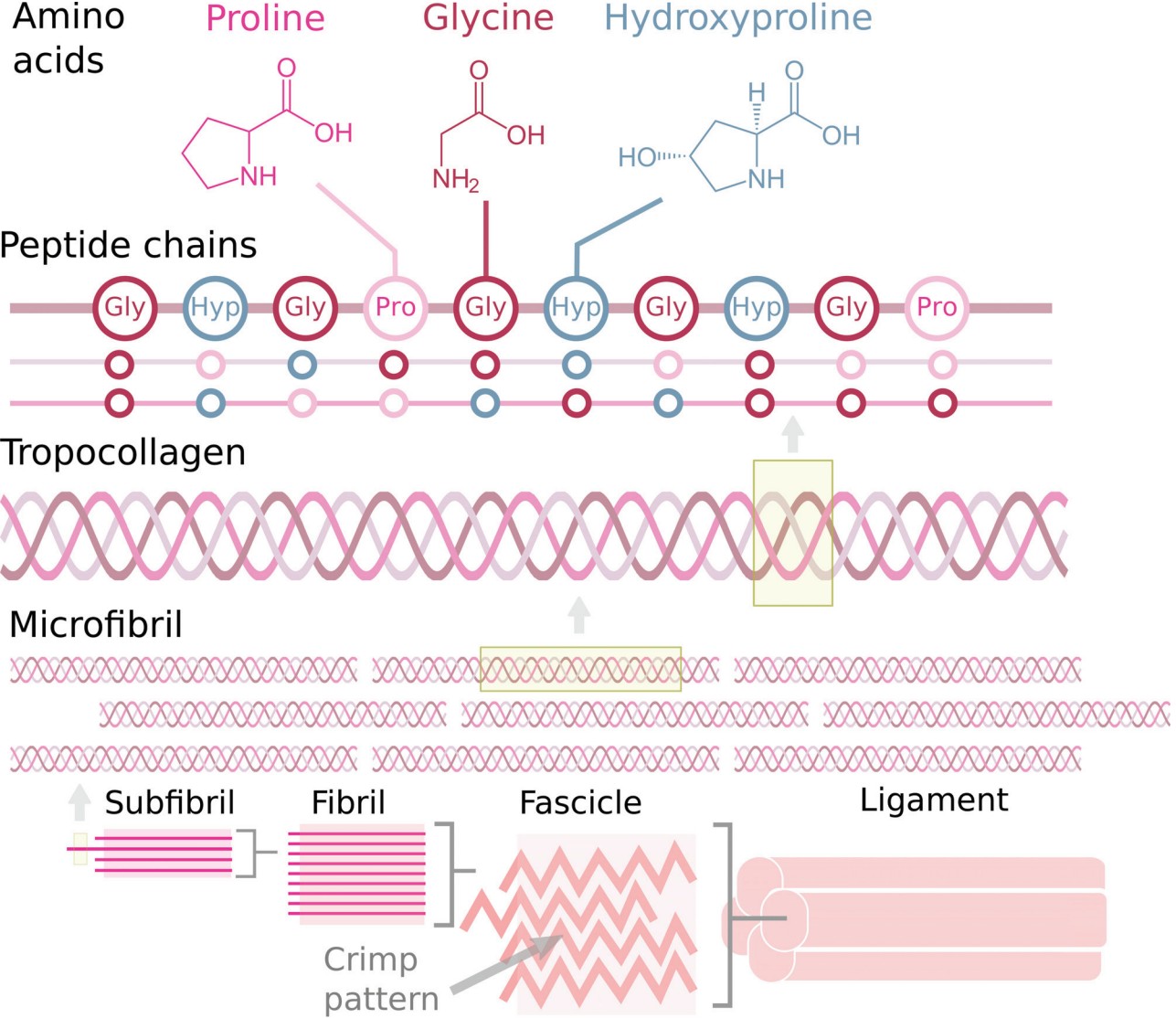

**Fig 1. Ligament structure.** The hierarchical structure of ligaments, and organization of the tropocollagen from triple-helical polypeptide chains.

Recently, NIRS has been suggested as a tool for real-time tissue diagnostics in orthopedic applications [24, 25]. The operating principle of NIRS-based diagnostics is to supplement traditional arthroscopic tissue evaluation methods (i.e., visual observation and manual palpation) with quantitative point measurements made with NIRS. These measurements can be made in real-time and provide more realistic assessment of tissue integrity. Furthermore, the NIRS probe used to perform these measurements can be manufactured in the same shape and size as the traditional arthroscopic hooks. Earlier, NIRS has successfully been used to evaluate mechanical and biochemical properties of connective tissues (e.g., articular cartilage [26], meniscus [27], and subchondral bone [28]). Even more recently, the effectiveness of NIRS-based evaluation of articular cartilage was demonstrated during *ex vivo* arthroscopy [29]. Related to NIRS-based evaluation of ligaments, only Padalkar et al. [30] and Torniainen et al. [31] have evaluated the water content of ACL and the mechanical properties of knee ligaments, respectively. However, comprehensive evaluation of the biochemical and structural properties

of knee ligaments with NIRS has not been previously attempted. As the chemical composition of ligaments is very similar to cartilage and meniscus (i.e., collagen and water), it is plausible that NIRS analysis of ligaments will yield similar findings. NIRS-based evaluation of the chemical constituents and structural properties of knee ligaments provides an attractive option for real-time tissue diagnostics in orthopedic applications. The proposed technique could be useful in determining the extent of damage or selecting suitable replacement tissue (e.g., ACL graft) in orthopedic repair surgeries.

We hypothesized, based on earlier studies of regarding NIRS-based evaluation of connective tissues, that NIRS is able to quantify biochemical (and possibly structural) properties of knee ligaments and the patellar tendon. In this study, NIRS was used to estimate the chemical composition and collagen crimp parameters of knee ligaments extracted from bovine stifle joints. Several NIR spectra were acquired from each sample which were then subjected to biochemical and histological analyses. Interested readers should know, that the same samples used in this study, have previously been analysed for biochemistry, biomechanical properties, and their relationship by Ristaniemi et al [8, 32]. These studies verified, for instance, the link between collagen content and the strength and toughness of the tissue, as well as the role of proteoglycans in modulating functional properties of ligaments. The relationship between NIRS and biomechanical properties of these samples was investigated by Torniainen et al [31]. The NIRS and reference data used in this study are also available as open access datasets [33]. To our knowledge, this is the first time prediction models have been developed to estimate the main chemical components and structural properties of the five primary knee ligaments from their NIR spectra.

As the overall health of knee ligaments is largely dictated by their chemical composition and structure, the ability to non-destructively evaluate these parameters could hold substantial diagnostic potential as an arthroscopic tool. A more quantitative approach to arthroscopic evaluation of tissue integrity could improve the quality of orthopedic treatment and patient outcomes. Since the capabilities of the NIRS technique for evaluating various properties of similar tissues (i.e., cartilage, meniscus, and bone) have been demonstrated, the addition of ligaments could yield a novel all-in-one quantitative diagnostic technique which (with minor modifications) would be suitable for evaluating all the connective tissues within the knee joint.

## Materials and methods

### Sample extraction

ACL, PCL, LCL, MCL, and PT were extracted from 10 healthy skeletally mature bovine stifle joints (Fig 2a). The joints were acquired from a slaughterhouse (Atria Oyj, Seinäjoki, Finland); thus, no ethical permission was required. Ligaments and tendons were stored frozen (-20˚C) immediately after extraction. The tissues were thawed 24 hours before spectroscopic measurements and kept in a refrigerator (4˚C) before starting the spectroscopic measurements. A small dumbbell-shaped sample piece (region of interest approximately $2.0 \times 1.8 \times 10$ mm in dimensions) was cut from the midsection of each ligament using a custom punch-tool (Fig 2b). Midsection was selected for the sample extraction site as the material properties of the tissue are relatively uniform at this location. The samples were prepared in this way to make them suitable for mechanical tensile testing which was performed prior to the biochemical and histological analyses [32].

### Near infrared spectroscopy

NIR spectra were measured using a commercially available benchtop NIRS system (AvaSpec Multichannel Spectrometer, Avantes BV, Apeldoorn, Netherlands), consisting of a light source

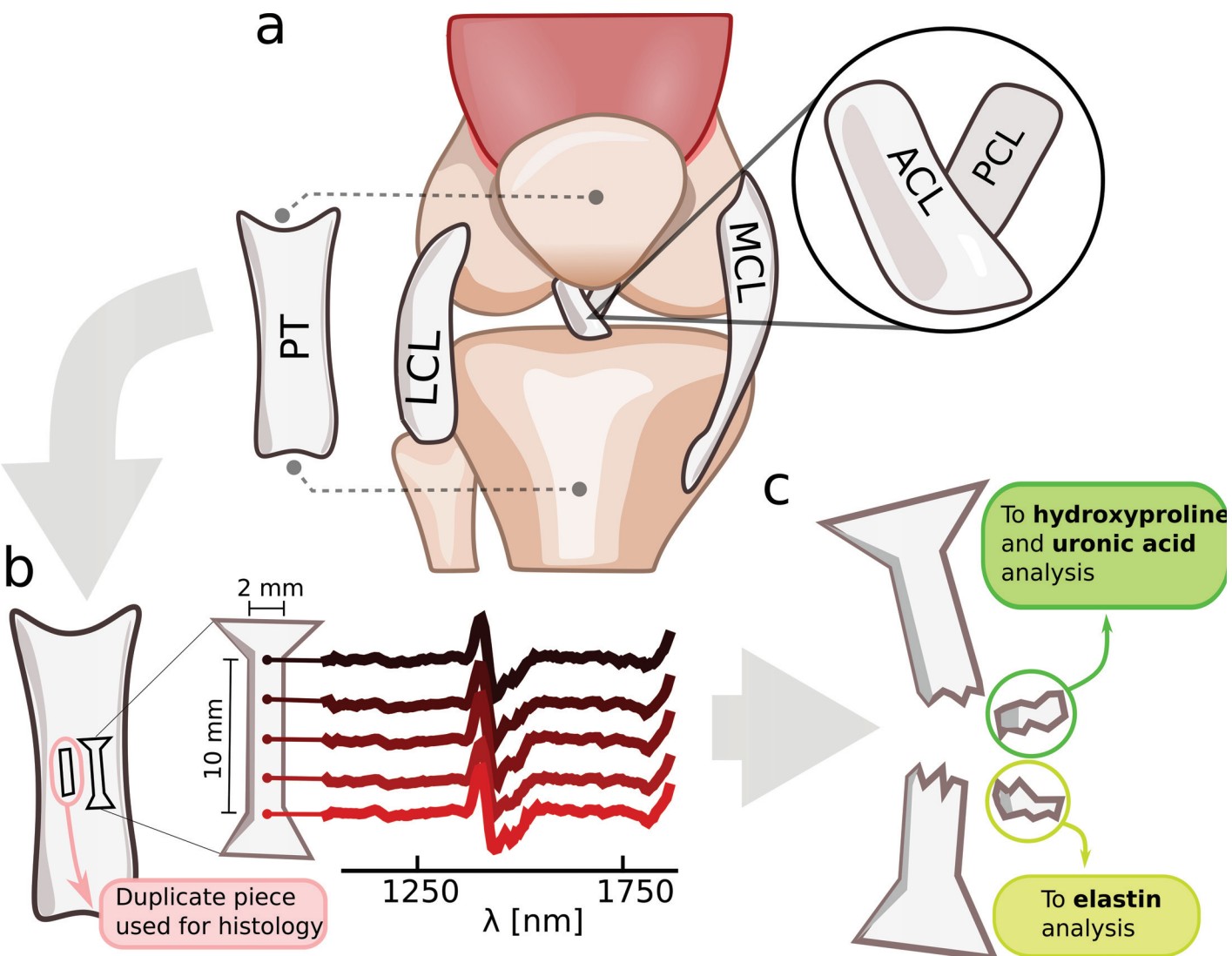

**Fig 2. Sample processing and measurement techniques. a**: Four knee ligaments and the patellar tendon were extracted from 10 bovine stifle joints (right human knee used here for visualization purposes). **b**: Full ligaments were cut into smaller sample pieces which were then measured using NIRS from five equispaced locations along the longitudinal axis. A duplicate piece of tissue adjacent to the sample piece was extracted for histological analysis which determined the length and angle of the collagen crimp. **c**: The ligament samples were subjected to destructive mechanical testing protocol (see [32] for details) which ruptured the samples. Pieces of the samples were then used for hydroxyproline, elastin, and uronic acid quantification.

(10W Tungsten Halogen Lamp), two detectors (Detector 1: AvaSpec-ULS2048L-USB2 for wavelength region 350–1100 nm and Detector 2: AvaSpec-NIR256-2.5-HSC for wavelength region 1000–2400 nm) and a custom-made probe shaped like an arthroscopic hook. The stainless-steel probe (outer diameter = 3.25 mm, diameter of the fiber bundle = 1.90 mm) contains 114 optical fibers (d = 100 $\mu$m) with 100 fibers emitting and 14 fibers (7 per detector) collecting light to the spectrometers. The probe has been specifically designed for *in vivo* arthroscopic measurement of connective tissues of the knee [28]. During acquisition, the measured spectrum consisted of 100 coadded scans with integration time of 1.5 milliseconds for AvaSpec-ULS2048L-USB2 detector and 20 milliseconds for AvaSpec-NIR256-2.5-HSC detector. NIR spectra (Fig 3a) were collected at five equispaced non-overlapping sites (2.0 mm spacing)

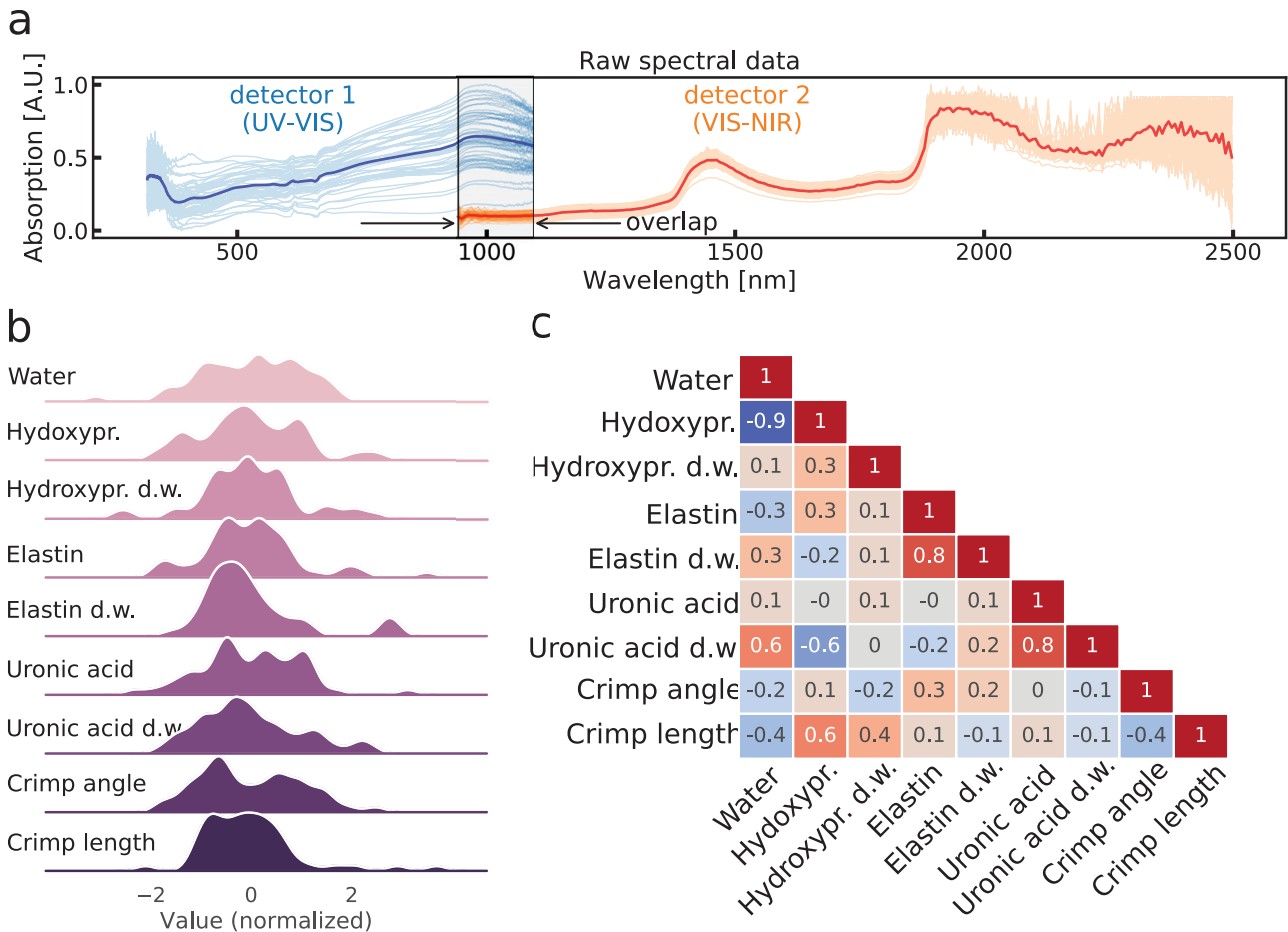

**Fig 3. NIRS measurements and reference variables. a**: Full wavelength range of the collected NIR spectra from detector 1 (blue) and detector 2 (orange). Thick lines represent the average spectra. Absorption values from both detectors have been rescaled to range from 1 to 0 to enable easier comparison. **b**: Normalized distribution of the reference properties. **c**: Correlation between the reference properties. Color of the boxes represents the sign and strength of the correlation.

along the longitudinal axis of the sample (Fig 2b). In the analysis, the five sites were averaged as the biochemical and structural properties were assumed uniform over the sample. Before each measurement, the system was calibrated using a diffuse reflectance standard (99% ± 4% reflectance factor in the 200–2500 nm range, Spectralon SRS-99, Labsphere Inc., North Sutton, USA) and a non-reflective standard. The non-reflective standard corresponds to the baseline noise measured in the absence of any incoming light. Conversely, the reflectance standard corresponds to the maximal signal-to-noise ratio the detector is capable of reaching without saturating the signal. Reference spectra were used to compute absorption values ($A$) for each individual wavelength ($\lambda$) using the equation,

$$A_\lambda = -log_{10}\left(\frac{S_\lambda - D_\lambda}{R_\lambda - D_\lambda}\right),\tag{1}$$

where $S_\lambda$ is the sample spectrum, $D_\lambda$ the spectrum of the non-reflective standard, and $R_\lambda$ the spectrum of the reflectance standard.

Tissue samples were held in place by a sample holder inside a petri dish mounted on top of a goniometer (#55-841, Edmund Optics Inc., Barrington, NJ, USA) which was attached to a

three-axis actuator system (ESP300 Motion Controller/Driver, Newport Corporation, Irvine, CA, USA and T-LSQ300B, Zaber Technologies Inc., Vancouver, British Columbia, Canada) for precise control of the sample position during measurements. Ambient light was minimised during each measurement. The sample orientation was adjusted to ensure perpendicular contact between the probe tip and the sample surface before the measurement. During the measurements, the samples were submerged in phosphate-buffered saline (PBS) to ensure sufficient hydration. Since PBS is a strong absorber of NIR light due to the high water content, special care was taken to ensure tight seal between the probe tip and the sample surface. The tight contact prevented excessive PBS from being registered by the probe and thus improved the quality of the measured spectra. Signal quality was confirmed visually before each measurement, as PBS contamination can easily be seen as spurious absorbance peaks.

## Biochemical and histological analysis

The biochemical analysis procedure was previously reported in Ristaniemi et al. [8] and is only briefly recapitulated here. Prior to biochemical and histological analyses, a mechanical testing protocol was performed on the samples (details of the protocol reported by Ristaniemi et al. [32]). During the biomechanical testing the samples were submerged in PBS to ensure proper hydration. The final part of the mechanical testing protocol was a quasi-static ultimate tensile test which resulted in a ruptured sample. Biochemical analyses were performed on small sample pieces (7–38 mg) extracted from the tissue after the mechanical testing (Fig 2b). The extracted testing pieces were stored frozen (-20˚C) for 5 months, followed by thawing and rehydration before the biochemical analysis. The total water content was determined by weighting the sample before and after drying it in a lyophilizer for 24 hours. Standard biochemical assays [8] were conducted to determine the wet- and dry-weight quantities of hydroxyproline (indicative of the total collagen content), uronic acid (indicative of the total proteoglycan content), and the total amount of elastin (Table 1). A tissue sample, adjacent to the dumbell-shaped sample, was extracted for structural analysis. A 5 $\mu m$ thick piece was imaged using a combination of polarized light imaging system (Abrio, CRi Inc., Woburn, MA, USA) and polarized light microscope (Nikon Diaphot TMD, Nikon Inc., Shingawa, Tokyo,

**Table 1. Distributions of the biochemical and structural reference variables.**

|  | Mean | SD | Min. | 25% | 50% | 75% | Max. |
|---|---|---|---|---|---|---|---|
| Water content [%] | 74.7 | 5.6 | 57.4 | 70.7 | 75.3 | 79.1 | 84.4 |
| Collagen content d.w. [%] | 90.2 | 8.1 | 69.0 | 85.1 | 89.8 | 94.5 | 110.2 |
| Elastin content d.w. [%] | 4.5 | 1.3 | 2.5 | 3.7 | 4.2 | 4.9 | 8.2 |
| Pg content d.w. [%] | 1.0 | 0.4 | 0.3 | 0.7 | 0.9 | 1.2 | 1.9 |
| Hydroxyproline w.w. [$\mu g/mg$] | 32.5 | 7.5 | 18.4 | 28.0 | 31.9 | 38.2 | 51.6 |
| Hydroxyproline d.w. [$\mu g/mg$] | 128.8 | 11.6 | 98.5 | 121.6 | 128.3 | 134.9 | 157.4 |
| Uronic acid w.w. [$\mu g/mg$] | 0.5 | 0.1 | 0.1 | 0.4 | 0.5 | 0.6 | 0.9 |
| Uronic acid d.w. [$\mu g/mg$] | 1.9 | 0.8 | 0.6 | 1.4 | 1.8 | 2.4 | 3.8 |
| Elastin w.w. [$\mu g/mg$] | 11.7 | 3.3 | 5.6 | 9.9 | 11.7 | 13.3 | 23.1 |
| Elastin d.w. [$\mu g/mg$] | 45.5 | 12.8 | 25.2 | 37.1 | 42.5 | 48.9 | 81.6 |
| Crimp angle [˚] | 27.1 | 6.5 | 15.9 | 22.4 | 26.4 | 32.2 | 43.0 |
| Crimp length [$\mu m$] | 57.1 | 15.7 | 24.4 | 45.8 | 54.7 | 62.6 | 114.6 |

The total amounts of collagen and proteoglycans was estimated from hydroxyproline and uronic acid contents, respectively. Estimates provided for both wet weight (w. w.) and dry weight (d.w.).

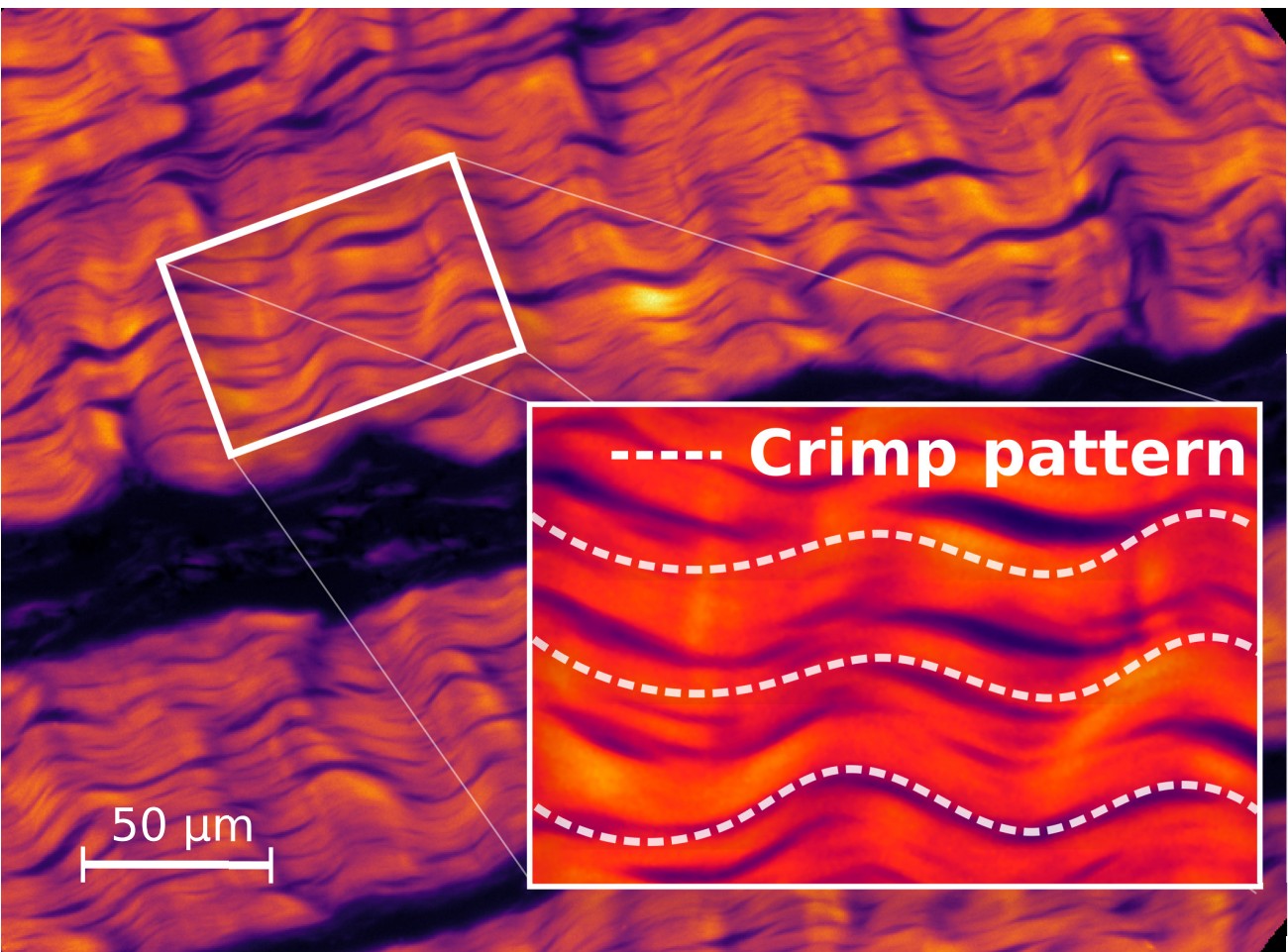

**Fig 4. Polarized light microscopy.** An example optical retardance image of ACL sample used to determine the crimp parameters. The values for crimp length and angle were calculated as an average from manually selected regions of interest.

Japan). Parameters describing the length and angle of collagen fiber crimping were computed from manually selected regions of interest (ROIs) from the resulting images [34] (Fig 4).

## Statistical analysis

Each biochemical and structural reference variable was analysed individually by building a corresponding NIRS prediction model using PLSR. Prior to model construction, each reference variable was standardized. Model construction consisted of preprocessing, variable selection, and regression. Final performance of the model was evaluated using median performance metrics of 10-fold cross-validation which was repeated 10. The performance metrics included explained variance (Eq 2), Pearson correlation coefficient ($r_{CV}$), coefficient of determination ($R2_{CV}$), and root mean squared error ($RMSE_{CV}$).

$$explained\_variance(y, \hat{y}) = 1 - \frac{Var\{y - \hat{y}\}}{Var\{y\}} \qquad (2)$$

Specifically, each model was constructed using the following steps:

1. Numerous (N = 96) NIRS preprocessing options were tested using the open-source `nippy` Python module [35] (see section NIRS Preprocessing for details).

2. To reduce the number of variables, 10% of the wavelengths with the lowest variance were rejected.

3. Number of variables was further reduced by selecting the top 87.5% of the remaining wavelengths in terms of univariate regression testing.

4. The number of latent variables (LVs) for PLS was determined using 5-fold cross-validation

5. Number of variables was further reduced using backwards feature elimination in combination with PLSR with a cut-off point set to the number of LVs determined in the previous step.

6. The final model was evaluated using a 10-fold cross-validation, repeated 10 times with randomized splits. The performance metrics for the model were obtained from the median performance over all the folds with bootstrapped confidence intervals.

The best performing model was selected for each biochemical and structural variables from all the tested models. Nearly all of the dependent biochemical and structural variables were normally distributed (Fig 3b). A skewness and kurtosis based test of normality only flagged elastin content (dry-weight) and crimp length as not being normally distributed. Correlation between reference variables have been listed in Fig 3c. Each sample was considered independent and ligament type was not included in the model. All analysis was performed in Python using `numpy` [36], `SciPy` [37], `scikit-learn` [38], and `nippy` [35] packages.

**NIRS preprocessing.**   As the instrumentation consisted of two separate detectors, the resulting spectra were preprocessed separately, resampled to same spectral resolution, and then concatenated. All PLSR models used the concatenated spectrum as the independent variable.

Preprocessing protocols for spectra from detector 1 (350–1100 nm) included

- 3rd order Savitzky-Golay filtering at filter window ranges: 17.7–86.1 nm

- Computation of spectral derivatives up to the 2nd degree.

- Standard normal-variate scatter correction.

- Limiting the spectra to wavelength ranges of 370–980 nm, 675–960 nm, and 430–680 nm.

  Preprocessing protocols for spectra from detector 2 (1000–2400 nm) included

- 3rd order Savitzky-Golay filtering at filter window ranges: 57.7–198.7 nm

- Computation of spectral derivatives up to the 2nd degree.

- Standard normal-variate scatter correction.

- Limiting the spectra to wavelength ranges of 1000–1900 nm, 900–1380 nm, and 1550–1870 nm.

Specific wavelength bands were selected to exclude regions with particularly noisy or saturated signal.

Optimization of the preprocessing techniques for the NIRS models predicting water and hydroxyproline content converged to the same preprocessing pipeline. The preprocessing pipeline for the spectra from detector 1 consisted of a Savitzky-Golay filtering that produced a smoothed 2nd derivative of the spectra by using a 3rd degree polynomial fit with a 18 nm

**Table 2. Median cross-validated correlation ($r_{CV}$), R2$_{CV}$, and root mean squared error ($RMSE_{CV}$) of different reference properties.**

| Property | Wavelengths | LVs | $r_{CV}$ | $R2_{CV}$ | $RMSE_{CV}$ |
|---|---|---|---|---|---|
| Water content | 161 / 232 | 13 | 0.89 (0.87–0.92) | 0.65 (0.53–0.71) | 0.48 (0.41–0.52) |
| Hydroxyproline content | 159 / 232 | 11 | 0.88 (0.84–0.90) | 0.57 (0.48–0.68) | 0.56 (0.50–0.60) |
| Hydroxyproline content d.w. | 90 / 232 | 2 | 0.57 (0.44–0.73) | 0.06 (-0.23–0.21) | 0.85 (0.71–0.92) |
| Uronic acid content | 153 / 364 | 1 | 0.52 (0.39–0.60) | 0.05 (-0.10–0.13) | 0.84 (0.80–0.87) |
| Uronic acid contet d.w. | 171 / 232 | 3 | 0.66 (0.55–0.73) | 0.14 (-0.30–0.26) | 0.86 (0.76–0.91) |
| Elastin content | 74 / 232 | 1 | 0.48 (0.27–0.63) | -0.19 (-0.35–-0.06) | 0.82 (0.73–0.90) |
| Elastin content d.w. | 56 / 88 | 4 | -0.01 (-0.05–0.20) | -0.53 (-0.83–-0.35) | 0.87 (0.78–1.03) |
| Crimp angle | 46 / 88 | 4 | 0.62 (0.54–0.67) | 0.04 (-0.11–0.20) | 0.80 (0.76–0.85) |
| Crimp length | 11 / 364 | 2 | 0.37 (0.26–0.43) | -0.33 (-0.49–-0.16) | 0.80 (0.73–1.00) |

Reference properties have been standardized to enable easier comparison. Bootstrapped confidence interval of each metric is given inside parentheses. The number of latent variables (LVs) used for PLSR models is also provided. The effect of variable selection is reported as the ratio of chosen wavelengths to the full range of initial wavelengths after preprocessing.

filtering window. After the filtering operation, the spectra was limited to the wavelength range of 370–980 nm. No additional scatter correction was applied to the spectra. The spectra from detector 2 was similarly preprocessed with a Savitzky-Golay filter that produced a smoothed 2nd derivative of the spectra by using a 3rd degree polynomial fit with 58 nm filtering window. With detector 2, the spectra was limited to the wavelength range of 1000–1900 nm. Again, no normalization or scatter correction operations were applied to the spectra.

## Results

The PLSR models predicting biochemical reference variables (Table 2) produced the highest cross-validated accuracy (in terms of variance explained, Fig 5) in evaluating water (73%) and hydroxyproline content (68%). These models decomposed the input spectra to 13 and 11 latent variables, respectively. Prediction of crimp angle yielded the highest prediction performance (31%) amongst the structural variables. All remaining models performed poorly in terms of prediction accuracy ($< 32\%$). PLSR models exceeding 50% in terms of variance explained, were subjected to further analysis. Analysis of the largest regression coefficients (Fig 6a) indicated that while the hydrogen-oxygen bonds of water clearly contribute to the model, it is also affected by NIR absorption of some collagen constituents (i.e., CH-groups), resulting from hydroxyproline, glycine, or proline. The preprocessed spectra used for predicting water and hydroxyproline contents is visualized in Fig 6b.

## Discussion

In this study, the ability of NIRS to evaluate the chemical composition and internal structure of ligaments was investigated. Although some properties of ligaments have been investigated in the past with NIRS, comprehensive study of all major chemical and structural properties of the five main knee ligaments has not been previously done. Determining the ability of NIRS to estimate composition and crimp properties of ligaments is an important step towards a comprehensive arthroscopic NIRS characterization of connective tissues of the knee.

Of all the constructed PLSR models, the highest accuracy was observed with water and hydroxyproline contents; although, both of these reference variables were highly correlated (Fig 3c). Investigation of model coefficients revealed water and organic molecules (i.e., hydroxyproline, glycine or proline) to be the main contributors to the PLSR models. For the

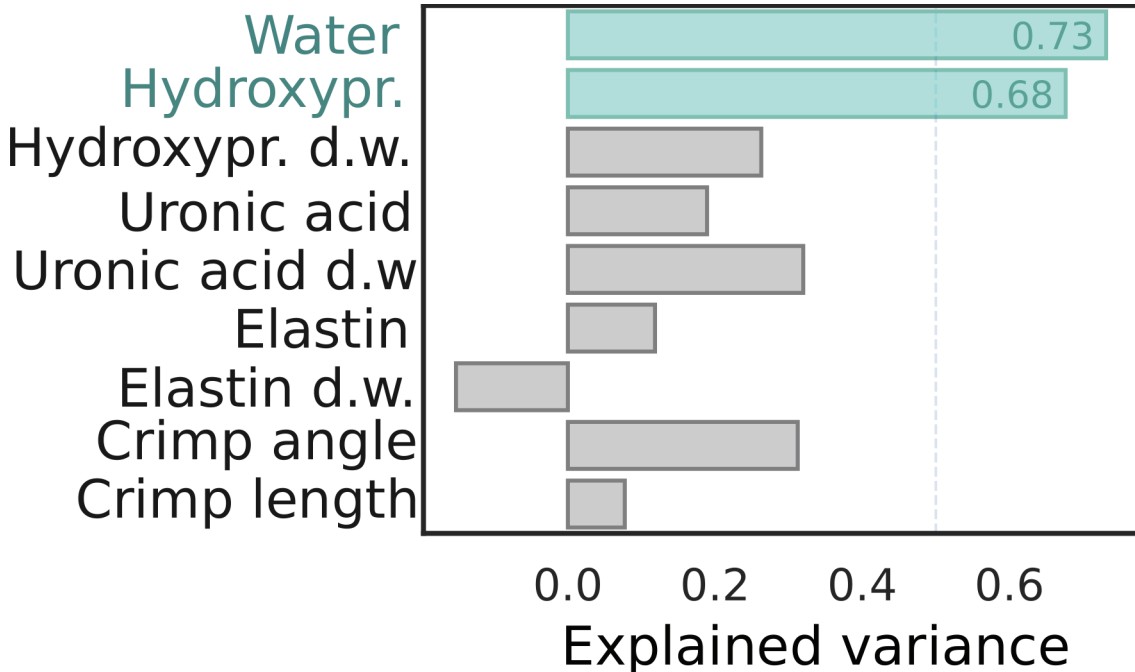

**Fig 5. Accuracy of the predictive models.** Median prediction performance (in terms of explained variance) for nine different biochemical and structural properties. Median performance was computed using a 10-fold repeated (N = 10) cross-validation. Only the prediction for water and hydroxyproline content (of wet weight) produced models with over 50% of the variance explained.

model predicting the water content, the five most influential coefficients (in terms of absolute magnitude) were attributed to wavelengths of 552, 852, 1178, 1224, and 1893 nm. Corresponding wavelengths for the model predicting hydroxpyproline were 1224, 1643, 1682, 1688, 1874 nm. NIR absorption of water is most prominent around wavelengths of 760, 970, 1190, 1450, and 1940 nm [39]. In the model predicting water, two high magnitude coefficients are located close to water absorption peaks (i.e., within 50 nm) while the model for hydroxpyroline content only contains one such coefficient. Additionally, both models assigned high coefficients to wavelengths of 552 and 1224 nm.

The question whether the models are capable of predicting water or collagen content is academic, as the high correlation between the two means that knowing one quantity automatically reveals the other. In other words, NIRS is capable of estimating the amount of collagen in bovine knee ligaments. As the total amount of collagen largely dictates the mechanical behaviour of ligaments, the sensitivity of NIRS towards collagen can be considered as the most important indicator of the technique's diagnostic potential. Because ligaments are essentially mechanical tissues, any deviation in their collagen-mediated mechanical response may be linked with ligament health. Quantitative tissue evaluation during arthroscopy could provide additional diagnostic information during orthopedic repair procedures or help to select optimal rehabilitation procedures after the operation. As this technique is invasive, it should only rarely be performed solely for the diagnosis of injury. Rather, the technique can provide vital supplemental information about the tissue condition when surgical intervention has been decided as treatment.

An earlier investigation on NIRS and mechanical properties of bovine stifle joint ligaments studied the potential of NIRS to predict the tissue's response to sinusoidal, stress-relaxation, and quasi-static loading [31]. Of the three loading types, only the quasi-static properties could

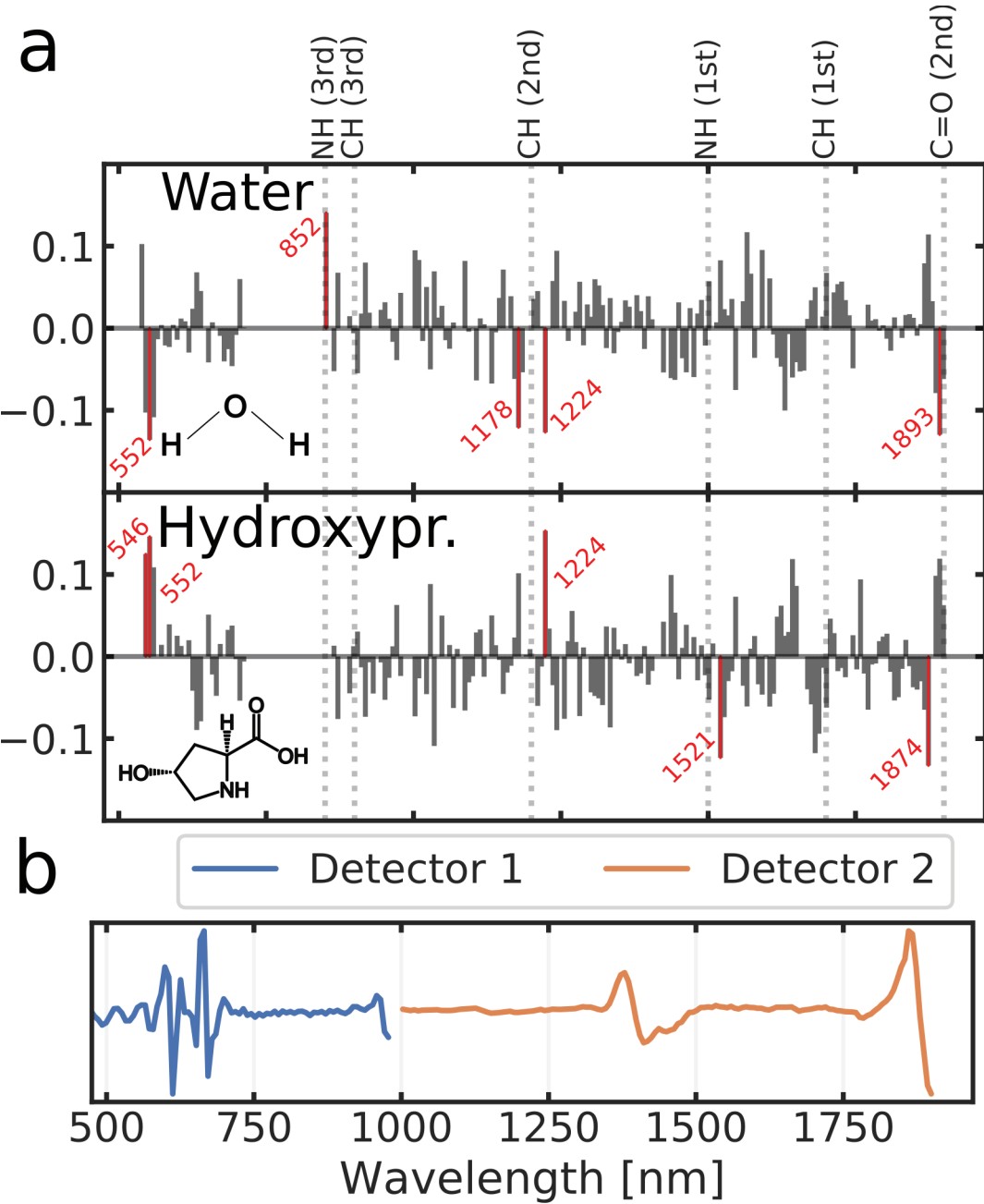

**Fig 6. Feature importance of the predictive models. a**: Coefficients of the PLS models for water and hydroxyproline content. The five most influential wavelengths (in terms of absolute magnitude) have been highlighted in red. **b**: Spectrum used as input for the water and hydroxyproline models. Spectra from detector 1 was smoothed with a 3rd order polynomial Savitzky-Golay filter using a 18 nm filtering window and the second spectral derivative. Similar treatment was applied to spectra from detector 2 with the exception of filtering window being set to 58 nm. Spectra from detector 1 was limited to the wavelength range of 370–980 nm and the spectra from detector 2 to 1000–1900 nm.

be predicted with NIRS. In quasi-static loading, the ligament was slowly stretched until the failure point with the resulting stress-strain curve describing toe region, linear region, yield, and failure characteristics of the sample. The properties related to the yield and failure point of the ligament could be predicted with higher accuracy than the other mechanical properties.

Quasi-static properties are mainly linked to un-crimping, stretching, and failure of collagen structures within the ligament. The total amount of collagen, therefore, has a substantial contribution to quasi-static response of the tissue. It is likely that the reported results were also indirectly based on detecting collagen content. Padalkar et al. [30] previously reported that NIRS is sensitive towards the water content in ACLs. Similar effect was also observed here, as evident from the explained variance of the PLSR model when predicting water content.

NIRS-based estimation of elastin and proteoglycan contents of the ligaments exhibited much lower accuracy than water and hydroxyproline. This was to be expected as the total quantity and variation of these contents is unlikely to exceed the detection limit of the used NIRS instrumentation (Table 1). Likewise, the two crimp parameters (length and angle) describing the organization of the collagen fascicles were predicted with poor accuracy. It appears that the changes in the collagen crimp do not sufficiently alter the scattering of NIR light, and therefore, NIRS is unable to characterize this property of ligaments. This finding is consistent with the previously reported poor performance [31] in predicting the non-linear toe region behaviour of ligaments which is believed to be defined by the collagen crimp.

Even though the biochemical composition [8] and collagen crimp parameters [40] differ between primary ligaments and PT, ligament type was not included as a predictor in the PLSR models. Rather, each sample was treated as independent ligamentous tissue. This decision was intentional as the main object of the study was to evaluate the capability of NIRS to predict biochemical and structural properties directly. Since all the reference variables were explicitly known for each sample, differentiating them by anatomical location was deemed unecessary. We would like to point out, however, that the ligaments were previously analysed separately by Ristaniemi et al. for biomechanics and biochemistry [8, 32].

The total number of samples used in this study was relatively small and, therefore, the models might not be entirely representative of real world performance. This was countered by the means of repeated cross-validation techniques when evaluating the model accuracy. Additionally, the pool of samples exhibited limited biological variability as the samples were extracted from animals of similar health and age. However, bovine stifle joint ACL and PCL are generally considered as a good animal-models of human ligaments in terms of size and anatomy [41]. The determination of collagen crimp was done from a single ROI (per sample) which might have induced some error in the structural models, if the distribution of crimp parameters are not uniformly distributed along the ligament mid-section. Finally, the time each sample spent submerged in PBS could have affected the water content of the tissue. This should not, however, affect the predictive performance of the NIRS model as each sample was treated in an identical fashion (i.e., the net change in water content should be approximately the same for all samples).

## Conclusion

In conclusion, the findings of this study corroborate earlier speculations of NIRS being able to detect bulk collagen content of knee ligaments [31]. The accuracy of NIRS-based evaluation of ligament composition is roughly comparable to the accuracy of similar applications used to evaluate other collagenous tissues [27, 30]. Arthroscopic NIRS could, therefore, also be used for ligament diagnostics alongside other connective tissues of the knee. Additional validation through *in vivo* studies, however, would still be desirable.

## Acknowledgments

Mondal, D. is acknowledged for acquisition and analysis of polarized light microscopy images.

## Author Contributions

**Conceptualization:** Lauri Stenroth, Juha Töyräs.

**Data curation:** Jari Torniainen, Aapo Ristaniemi.

**Formal analysis:** Jari Torniainen, Jaakko K. Sarin.

**Investigation:** Jari Torniainen, Aapo Ristaniemi, Lauri Stenroth.

**Methodology:** Jari Torniainen, Aapo Ristaniemi, Jaakko K. Sarin, Mithilesh Prakash, Isaac O. Afara, Mikko A. J. Finnilä.

**Project administration:** Juha Töyräs.

**Resources:** Mikko A. J. Finnilä, Rami K. Korhonen, Juha Töyräs.

**Supervision:** Lauri Stenroth, Rami K. Korhonen, Juha Töyräs.

**Validation:** Jaakko K. Sarin, Mithilesh Prakash, Isaac O. Afara, Lauri Stenroth.

**Visualization:** Jari Torniainen.

**Writing – original draft:** Jari Torniainen.

**Writing – review & editing:** Aapo Ristaniemi, Jaakko K. Sarin, Mithilesh Prakash, Isaac O. Afara, Mikko A. J. Finnilä, Lauri Stenroth, Rami K. Korhonen, Juha Töyräs.

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
