## [Decision Letter · Decision Letter 0]

19 Oct 2021

PONE-D-21-18102Near Infrared Spectroscopic Evaluation of Biochemical and Crimp Properties of Knee Joint Ligaments and Patellar TendonPLOS ONE

Dear Dr. Torniainen,

Thank you for submitting your manuscript to PLOS ONE. After careful consideration, we feel that it has merit but does not fully meet PLOS ONE’s publication criteria as it currently stands. Therefore, we invite you to submit a revised version of the manuscript that addresses the points raised during the review process.

Two experts in the field have carefully reviewed the manuscript entitled, "Near Infrared Spectroscopic Evaluation of Biochemical and Crimp Properties of Knee Joint Ligaments and Patellar Tendon". Their comments apre appended below.

Both reviewers are very much interested in the study.

The reviewer #1 acknowledged the findings obtained by this study with leaving several concerns. This reviewer felt very regretful  that a lot of data described in the methods they are nof fully reflected in the results, if these are satisfactory analyzed, the manuscript could have greater significance. 

The reviewer #2 pointed out several critical concerns which need clarification before publication.

We look forward to receiving your revised manuscript.

Kind regards,

Manabu Sakakibara, Ph.D.

Academic Editor

PLOS ONE

Journal Requirements:

2. Please include information on the slaughterhouse used, such as name and location.

“This research received funding from the following sources: SCITECO Doctoral Programme of University of Eastern Finland, Kuopio University Hospital (VTR project 5203111), the Academy of Finland (projects 286526, 324529, and 315820) and the Sigrid Juselius Foundation. Mondal, D. is acknowledged for measurement and analysis of polarized light microscopy imaging.

“JTo: Doctoral Programme in Science, Technology and Computing (SICTECO)

https://www.uef.fi/en/career-advancement#paragraph-3088

5203111 Valtion tutkimusrahoitus

https://www.psshp.fi/tutkimus/tutkimusrahoitus/valtion-tutkimusrahoitus.The funders had no role in study design, data collection and analysis, decision to publish, or preparation of the manuscript.

AR:286526 and 324529.Academy of Finland

https://www.aka.fi/en/The funders had no role in study design, data collection and analysis, decision to publish, or preparation of the manuscript.

Sigrid Juselius Foundation . https://www.sigridjuselius.fi/en/The funders had no role in study design, data collection and analysis, decision to publish, or preparation of the manuscript.

IA:315820; Academy of Finland

https://www.aka.fi/en/.The funders had no role in study design, data collection and analysis, decision to publish, or preparation of the manuscript.

RK:286526 and 324529;Academy of Finland

https://www.aka.fi/en/The funders had no role in study design, data collection and analysis, decision to publish, or preparation of the manuscript.

Sigrid Juselius Foundation

https://www.sigridjuselius.fi/en/The funders had no role in study design, data collection and analysis, decision to publish, or preparation of the manuscript.

4. Please update your submission to use the PLOS LaTeX template. The template and more information on our requirements for LaTeX submissions can be found at http://journals.plos.org/plosone/s/latex.

Reviewers' comments:

Reviewer's Responses to Questions

**Comments to the Author**

1. Is the manuscript technically sound, and do the data support the conclusions?

Reviewer #1: Yes

Reviewer #2: Yes

2. Has the statistical analysis been performed appropriately and rigorously? 

Reviewer #1: Yes

Reviewer #2: N/A

3. Have the authors made all data underlying the findings in their manuscript fully available?

Reviewer #1: Yes

Reviewer #2: Yes

4. Is the manuscript presented in an intelligible fashion and written in standard English?

Reviewer #1: Yes

Reviewer #2: Yes

5. Review Comments to the Author

Reviewer #1: This is a clearly written manuscript that investigates whether the technique of near infrared (NIR) spectroscopy will be useful for the non-destructive assessment of ligament and tendon composition, in particular in an arthroscopic setting via fiber optic. Four different bovine ligaments, and the patellar tendon, were investigated. A significant amount of data were collected, and I believe that there are interesting results that could be teased out, in addition to what was presented. Specifically, data tables that describe the differences in composition for the different ligaments and the patellar tendons would be interesting to see and should be added, as patellar tendon is often used to repair ligaments. The tensile testing results should be included, as well as correlations with water and collagen, and the other parameters. It would be most interesting if any of the NIR data predicted tensile strength. In addition, the crimp data is novel, and if differences were found among ligaments and PT, these should be discussed. Could a combination of any of the biochemical/structural parameters predict tensile strength via a multiple regression? In addition to adding in the additional data and analyses, some clarifications in methods and results would also be useful, as described below.

Comments:

1. Can the authors comment on the fact that the samples were submerged in PBS during NIR data collection ? How might this affect the spectra?

2. When were the water content measurements made? How were the tissues handled before and after the NIR measurements to ensure no changes to water content?

3. The last paragraph of results, lines 238 – 248, could be in Methods, not results.

4. Why are the tensile test results not included? These are important for understanding the rest of the data, and how differences in composition contribute to strength.

5. Page 13, line 261 paragraph, add in references for assignments of absorbances

Reviewer #2: This is an interesting study predicting the chemical compositions and structural properties of healthy bovine knee ligaments using NIRS in combination with chemometrics. The following queries need to be addressed:

1. Please add or emphasize the following information more clearly in the corresponding section: (1) sample fixation method before spectroscopic evaluation (formalin, PBS, cryofixation, etc); (2) storage period/time from sample extraction to spectroscopic evaluation; (3) their influence on spectral evaluation.

2. Were the spectra obtained from surface or couple of µm/mm inside or all depth of the samples? The results could be strongly affected by the light penetration depth. How did the authors consider that the probe geometry is suitable for assessing the ligament tissues?

3. Is there a consistent landmark that was used to ensure the measurement region of each sample? The ligaments are very heterogeneous, particularly while moving proximal to distal and between bundles so clarification should be made detailing your data sampling method and if it was not standardized it should be mentioned as a limitation in the discussion. Also, why did the author design the experiment without differentiating among ACL (AM/PL bundle), PCL (AL/PM bundle), LCL, MCL and PT despite their different functions and probably chemical compositions/ structural properties? The authors need to clearly justify (with either additional data or reference to the literature) why their differentiation was not made in this study.

4. Granted this paper was submitted to a non-clinical journal, however, you frequently mention that NIRS-based evaluation can be used for diagnosis in orthopedic repair surgeries. Can you please elaborate on this? The approach is certainly non-destructive (for the sample) but not non-invasive for patients. For example, clinically if you have a positive Lachman and pivot tests, most surgeons will go ahead and reconstruct the ACL regardless of whether the MRI actually shows a partial or complete rupture, thus, are you hoping to convince clinicians and medical device boards to allow an arthroscopic procedure prior to surgery to identify what state the ligament is in?

6. PLOS authors have the option to publish the peer review history of their article (what does this mean?). If published, this will include your full peer review and any attached files.

Reviewer #1: No

Reviewer #2: No

---

## [Author Response · Author response to Decision Letter 0]

22 Dec 2021

NOTE: ALL THIS INFORMATION IS CONVENIENTLY AND NICELY FORMATTED IN THE "Response to Reviewers.pdf" DOCUMENT. WE KINDLY RECOMMEND YOU READ THAT ONE INSTEAD.

We would like to thank the Editor and the Reviewers for their efforts and for providing us this opportunity to improve our manuscript. We have now compiled a point-by-point response to the issues raised by the Reviewers and also listed the implemented changes to the manuscript. We feel the changes proposed by the Reviewers have substantially increased the readability and scientific validity of the study.

EDITOR

------

Editor comment 1:

Authors' actions:

Manuscript now conforms to all of the PLOS ONE's style requirements.

Editor comment 2:

Please include information on the slaughterhouse used, such as name and location.

Authors' actions:

Slaughterhouse details have been added to the manuscript.

[Page 4, lines 111--112]: The joints were acquired from a slaughterhouse (Atria Oyj, Seinäjoki, Finland); thus, no ethical permission was required.

Editor comment 3

Please remove any funding-related text from the manuscript and let us know how you would like to update your Funding Statement. 

Authors' actions:

We have now removed all funding related text from the manuscript.

We have revised the statements and will provide them within the cover letter. Funding information is also listed here for the sake of completeness. We would also like to point out that Mithilesh Prakash has requested acknowledging Academy of Finland project 316258 as a funding source, which was not included in the original submission. 

SCITECO Doctoral Programme of University of Eastern Finland: Jari Torniainen

Valtion tutkimusrahoitus, Kuopio University Hospital, 5203111: Jari Torniainen

Academy of Finland, 324529: Rami K. Korhonen, Aapo Ristaniemi

Academy of Finland, 315820: Isaac O. Afara

Academy of Finland, 316258: Mithlesh Prakash *NEW*

Sigrid Juselius Foundation: Rami K. Korhonen, Aapo Ristaniemi

Editor comment 4

Please update your submission to use the PLOS LaTeX template. The template and more information on our requirements for LaTeX submissions can be found at http://journals.plos.org/plosone/s/latex

Authors' actions:

We have now updated our manuscript to use the PLOS LaTeX template.

REVIEWER 1

----------

Reviewer's summary:

This is a clearly written manuscript that investigates whether the technique of near infrared (NIR) spectroscopy will be useful for the non-destructive assessment of ligament and tendon composition, in particular in an arthroscopic setting via fiber optic. Four different bovine ligaments, and the patellar tendon, were investigated. A significant amount of data were collected, and I believe that there are interesting results that could be teased out, in addition to what was presented. Specifically, data tables that describe the differences in composition for the different ligaments and the patellar tendons would be interesting to see and should be added, as patellar tendon is often used to repair ligaments. The tensile testing results should be included, as well as correlations with water and collagen, and the other parameters. It would be most interesting if any of the NIR data predicted tensile strength. In addition, the crimp data is novel, and if differences were found among ligaments and PT, these should be discussed. Could a combination of any of the biochemical/structural parameters predict tensile strength via a multiple regression? In addition to adding in the additional data and analyses, some clarifications in methods and results would also be useful, as described below.

Authors' response:

We thank the Reviewer for these insightful comments. The interaction between biochemical composition and biomechanical properties of the ligaments is indeed of great interest to many researchers of musculoskeletal tissues. These results were, however, reported previously in two separate publications, namely

Ristaniemi, A., et al. "Comparison of elastic, viscoelastic and failure tensile material properties of knee ligaments and patellar tendon." Journal of Biomechanics 79 (2018): 31-38.

Ristaniemi, A., et al. "Comparison of water, hydroxyproline, uronic acid and elastin contents of bovine knee ligaments and patellar tendon and their relationships with biomechanical properties." Journal of the Mechanical Behavior of Biomedical Materials 104 (2020): 103639.

The latter also contains regression models describing the relationships between the biochemical and biomechanical properties. We would also like to remind that the relationship between NIRS and biomechanics was also investigated in

Torniainen, Jari, et al. "Near infrared spectroscopic evaluation of ligament and tendon biomechanical properties." Annals of Biomedical Engineering 47.1 (2019): 213-222.

Finally, the data used in this study has been released as an open access data paper

Ristaniemi, Aapo, et al. "Biomechanical, biochemical, and near infrared spectral data of bovine knee ligaments and patellar tendon." Data in Brief 36 (2021): 106976.

Authors' actions:

Earlier research is now highlighted in the Introduction section.

[Page 4, lines 88--95]: Interested readers should know, that the same samples used in this study, have previously been analysed for biochemistry, biomechanical properties, and their relationship by Ristaniemi et al\\cite{ristaniemi2018comparison, ristaniemi2020comparison}. These studies verified, for instance, the link between collagen content and the strength and toughness of the tissue, as well as the role of proteoglycans in modulating functional properties of ligaments. The relationship between NIRS and biomechanical properties of these samples was investigated by Torniainen et al\\cite{torniainen2019near}. The NIRS and reference data used in this study are also available as open access datasets\\cite{ristaniemi2021biomechanical}.

Reviewer comment 1:

Can the authors comment on the fact that the samples were submerged in PBS during NIR data collection? How might this affect the spectra?

Authors' response:

The Reviewer raises a valid concern regarding the potential interference caused by the PBS submersion in the NIRS measurements. As PBS is mostly water, it exhibits a very strong absorbance in the NIR region and could mask the absorbance spectrum of the tissue. In this study, the interference of PBS was minimized by ensuring a tight contact between the probe tip and the tissue surface. The measurement setup consisted of a goniometer mounted on top of a three-axis linear actuator stage which allowed for extremely precise control over the tissue-probe contact point. Before registering each measurement, the quality of the contact was verified visually from the spectrum, since even small amounts of PBS could easily be seen as saturation around the water peak of 1450 nm. With this setup, the effect of PBS on the spectrum come mainly from hydrating the tissue, which should be comparable to the natural hydration level of the ligaments.

Authors' actions:

We now mention the precautions taken against PBS interference when describing the NIRS measurements.

[Page 6, lines 155--161]: During the measurements, the samples were submerged in phosphate-buffered saline (PBS) to ensure sufficient hydration. Since PBS is a strong absorber of NIR light due to the high water content, special care was taken to ensure tight contact between the probe tip and the sample surface. The tight seal prevented excessive PBS from being registered by the probe and thus improved the quality of the measured spectra. Signal quality was confirmed visually before each measurement, as PBS contamination can easily be seen as a spurious absorbance peaks.

Reviewer comment 2:

When were the water content measurements made? How were the tissues handled before and after the NIR measurements to ensure no changes to water content?

Authors' response:

We appreciate the Reviewer's comment regarding handling of the sample and water content measurements.

First, the samples were submerged in PBS immediately after extraction from the stifle joints and then kept frozen (-20 °C) in the PBS solution until the measurements. Upon thawing, the samples were kept submerged in PBS for NIRS measurements and the following biomechanical testing. After the measurements, two smaller test pieces were separated from the tissue sample for biochemical analyses. These smaller samples were stored frozen (-20 °C) and rehydrated in PBS before the biochemical analysis (including water content determination). Biochemical analyses were performed approximately 5 months after NIRS and biomechanical measurements.

Constant submersion in PBS throughout the experiment and frozen storage during intervening times should have minimized the changes in water content for the tissue samples. The measurement times and sample handling procedures for each sample were near identical, thus any possible drying should affect all samples in a similar way and not interfere with the NIRS prediction accuracy.

Authors' actions:

We have supplemented the information regarding sample handling.

[Page 6, lines 163--172]: The biochemical analysis procedure was previously reported in Ristaniemi et al.\\cite{ristaniemi2020comparison} and is only briefly recapitulated here. Prior to biochemical and histological analyses, a mechanical testing protocol was performed on the samples (details of the protocol reported by Ristaniemi et al.\\cite{ristaniemi2018comparison}). During the biomechanical testing the samples were submerged in PBS to ensure proper hydration. The final part of the mechanical testing protocol was a quasi-static ultimate tensile test \\deleted{of the sample} which resulted in a ruptured sample. Biochemical analyses were performed on small sample pieces (7 -- 38 mg) extracted from the tissue after the mechanical testing (Fig. 2 b). The extracted testing pieces were stored frozen (-20 °C) for 5 months, followed by thawing and rehydration before the biochemical analysis.

Reviewer comment 3 [Results, lines 238 -- 248]:

The last paragraph of results, lines 238 – 248, could be in Methods, not results.

Authors' response:

We thank the Reviewer for pointing out this discrepancy and agree that the last paragraph is more suited for the Methods section.

Authors' actions:

We have now moved the last paragraph of Results section to the end of Methods section.

[Page 9, lines 255--256]: The preprocessed spectra used for predicting water and hydroxyproline contents is visualized in Fig. 6 b.

[Page 9, lines 257--268]: Optimization of the preprocessing techniques for the NIRS models predicting water and hydroxyproline content converged to the same preprocessing pipeline. The preprocessing pipeline for the spectra from detector 1 consisted of a Savitzky-Golay filtering that produced a smoothed 2nd derivative of the spectra by using a 3rd degree polynomial fit with a 18 nm filtering window. After the filtering operation, the spectra was limited to the wavelength range of 370 -- 980 nm. No additional scatter correction was applied to the spectra. The spectra from detector 2 was similarly preprocessed with a Savitzky-Golay filter that produced a smoothed 2nd derivative of the spectra by using a 3rd degree polynomial fit with 58 nm filtering window. With detector 2, the spectra was limited to the wavelength range of 1000 -- 1900 nm. Again, no normalization or scatter correction operations were applied to the spectra.

Reviewer comment 4:

Why are the tensile test results not included? These are important for understanding the rest of the data, and how differences in composition contribute to strength.

Authors' response:

We fully agree with the Reviewers comment about the importance of the tensile testing results. However, the results of the tensile testing protocol were previously reported in

Ristaniemi, A., et al. "Comparison of elastic, viscoelastic and failure tensile material properties of knee ligaments and patellar tendon." Journal of Biomechanics 79 (2018): 31-38.

The relationship between biochemical composition and biomechanics was investigated in

Ristaniemi, A., et al. ”Comparison of water, hydroxyproline, uronic acid and elastin contents of bovine knee ligaments

and patellar tendon and their relationships with biomechanical properties.” Journal of the Mechanical

Behavior of Biomedical Materials 104 (2020): 103639.

Likewise, the relationship between NIRS and the biomechanical properties were reported in

Torniainen, Jari, et al. "Near infrared spectroscopic evaluation of ligament and tendon biomechanical properties." Annals of Biomedical Engineering 47.1 (2019): 213-222.

The tensile test results and other biomechanical variables can also be found in the associated data paper.

Ristaniemi, Aapo, et al. "Biomechanical, biochemical, and near infrared spectral data of bovine knee ligaments and patellar tendon." Data in Brief 36 (2021): 106976.

We did not wish to duplicate the findings of these papers and rather just referred interested readers to the original publications. We understand that the existence of these publications and their relation to the current study were not expressed clearly enough.

Authors' actions:

We have now added the following passage to the Introduction to guide interested reader to the relevant papers.

[Page 4, lines 88--95]: Interested readers should know, that the same samples used in this study, have previously been analysed for biochemistry, biomechanical properties, and their relationship by Ristaniemi et al\\cite{ristaniemi2018comparison, ristaniemi2020comparison}. These studies verified, for instance, the link between collagen content and the strength and toughness of the tissue, as well as the role of proteoglycans in modulating functional properties of ligaments. The relationship between NIRS and biomechanical properties of these samples was investigated by Torniainen et al\\cite{torniainen2019near}. The NIRS and reference data used in this study are also available as open access datasets\\cite{ristaniemi2021biomechanical}.

Reviewer comment 5:

Page 13, line 261 paragraph, add in references for assignments of absorbances

Authors' response:}

We are grateful for the Reviewer to point out this missing yet crucial reference.

Authors' actions:

We have now added the most appropriate reference for the absorbance peak assignment (Curcio et al\\cite{curcio1951near}) and changed the exact values of the wavelengths accordingly.

[Pages 10--11, lines 285--288]: NIR absorption of water is most prominent around wavelengths of 760, 970, 1190, 1450, and 1940 nm\\cite{curcio1951near}. In the model predicting water, two high magnitude coefficients are located close to water absorption peaks (i.e., within 50 nm) while the model for hydroxpyroline content only contains one such coefficient.}

REVIEWER 2

----------

Reviewer's summary:

This is an interesting study predicting the chemical compositions and structural properties of healthy bovine knee ligaments using NIRS in combination with chemometrics. The following queries need to be addressed:

Authors' response:

We thank the Reviewer for their kind and encouraging summary of the manuscript.

Reviewer comment 1:

Please add or emphasize the following information more clearly in the corresponding section: (1) sample fixation method before spectroscopic evaluation (formalin, PBS, cryofixation, etc); (2) storage period/time from sample extraction to spectroscopic evaluation; (3) their influence on spectral evaluation.

Authors' response:

The Reviewer brings up valid points regarding the details of the study. 

1) Samples were submerged in PBS on a petri dish and held in place by a sample holder made from black rubber. The sample holder also doubled as a safety precaution to block any outside light, even though measurements were performed in a darkened room. The black rubber was the same material used as the dark reference when calibrating the spectrometer.

2) Ligaments and tendons were stored in PBS and frozen (-20 °C) immediately post extraction. The day before the actual measurements, the ligaments and tendons were first thawed, submerged in PBS, and stored in a refrigerator (4 °C) for approximately 24 hours before spectroscopic measurements. During these 24 hours, the ligaments/tendons were periodically taken out of the refrigerator and measured with various imaging modalities while keeping them in PBS at all times. These other measurements (ultrasound, scanning acoustic microscopy, and optical coherence tomography) were related to parallel studies and thus are not reported in this manuscript.

3) Samples were kept in PBS in order to maintain natural tissue hydration, which was vital for the accuracy and validity of the biochemical reference measurements. The duration from thawing to measurement could plausibly cause minor differences in the measured spectra. Careful consideration was taken, however, to ensure that each sample was measured in identical fashion. If the prolonged PBS exposure did affect the spectra, this effect would be similar for each sample. 

Authors' actions:

We have now revised the description of NIRS measurement setup to address the issues 1--3 raised by the Reviewer (and some issues raised by the other Reviewer).

[Page 4, lines 111--115]: The joints were acquired from a slaughterhouse (Atria Oyj, Seinäjoki, Finland); thus, no ethical permission was required. Ligaments and tendons were stored frozen (-20 °C) immediately after extraction. The tissues were thawed 24 hours before spectroscopic measurements and kept in a refrigerator (4 °C) before starting the spectroscopic measurements.

[Pages 5--6, lines 148--161]: Tissue samples were held in place by a sample holder inside a petri dish mounted on top of a goniometer (#55-841, Edmund Optics Inc., Barrington, NJ, USA) which was attached to a three-axis actuator system (ESP300 Motion Controller/Driver, Newport Corporation, Irvine, CA, USA and T-LSQ300B, Zaber Technologies Inc., Vancouver, British Columbia, Canada) for precise control of the sample position during measurements. Ambient light was minimised during each measurement. The sample orientation was adjusted to ensure perpendicular contact between the probe tip and the sample surface before the measurement. During the measurements, the samples were submerged in phosphate-buffered saline (PBS) to ensure sufficient hydration. Since PBS is a strong absorber of NIR light due to the high water content, special care was taken to ensure tight seal between the probe tip and the sample surface. The tight seal prevented excessive PBS from being registered by the probe and thus improved the quality of the measured spectra. Signal quality was confirmed visually before each measurement, as PBS contamination can easily be seen as spurious absorbance peaks.

[Page 12, lines 344--351]: The determination of collagen crimp was done from a single ROI (per sample) which might have induced some error in the structural models, if the distribution of crimp parameters are not uniformly distributed along the ligament mid-section. Finally, the time each sample spent submerged in PBS could have affected the water content of the tissue. This should not, however, affect the predictive performance of the NIRS model as each sample was treated in an identical fashion (i.e., the net change in water content should be approximately the same for all samples).

Reviewer comment 2:

Were the spectra obtained from surface or couple of µm/mm inside or all depth of the samples? The results could be strongly affected by the light penetration depth. How did the authors consider that the probe geometry is suitable for assessing the ligament tissues?

Authors' response:

We thank the Reviewer for focusing on this important aspect of light penetration in biological tissue.

The spectra were obtained from the surface of each sample piece with tight contact between the probe tip and the tissue surface. The measured signal consists of diffusely reflected NIR light with a penetration depth of up to a few millimeters, depending on the wavelength. Sample pieces themselves come from the mid-substance of each ligament/tendon. It is highly unlikely that there would be depth-wise variation in the biochemical composition as the properties in the mid-substance are rather homogeneous when compared to, e.g., the insertion sites\\cite{apostolakos2014enthesis}. Nevertheless, we estimate the penetration depth to be some millimeters into the tissue as similar penetration has been observed with articular cartilage\\cite{afara2021characterization}.

The probe has previously been used to successfully evaluate various properties of articular cartilage and meniscus. As a matter of fact, the design of the probe geometry was originally optimized to perform in vivo arthroscopic measurements. While several differences between ligaments, articular cartilage, and meniscus do exists, they share similar intrinsic characteristics. For instance, they all consists primarily of water and parallel collagen fibers. We, therefore, feel confident that this probe is more than adequate for performing NIRS measurements from ligaments and tendons.

Authors' actions:

We have now supplemented the description of sample extraction by highlighting that the samples were taken from the midsection.

[Page 4, lines 115--119]: A small dumbbell-shaped sample piece (region of interest approximately 2.0 x 1.8 x 10 mm in dimensions) was cut from the midsection of each ligament using a custom punch-tool (Fig. 2 b). Midsection was selected for the sample extraction site as the material properties of the tissue are relatively uniform at this location.

Suitability of the NIRS probe for musculoskeletal tissues is now also mentioned and we have included a new reference for interested readers.

Sarin, Jaakko K., et al. "Arthroscopic near infrared spectroscopy enables simultaneous quantitative evaluation of articular cartilage and subchondral bone in vivo." Scientific Reports 8.1 (2018): 1-10.

[Page 5, lines 131--132]: The probe has been specifically designed for in vivo arthroscopic measurement of connective tissues of the knee\\cite{sarin2018arthroscopic}.

Reviewer comment 3:

Is there a consistent landmark that was used to ensure the measurement region of each sample? The ligaments are very heterogeneous, particularly while moving proximal to distal and between bundles so clarification should be made detailing your data sampling method and if it was not standardized it should be mentioned as a limitation in the discussion. Also, why did the author design the experiment without differentiating among ACL (AM/PL bundle), PCL (AL/PM bundle), LCL, MCL and PT despite their different functions and probably chemical compositions/ structural properties? The authors need to clearly justify (with either additional data or reference to the literature) why their differentiation was not made in this study.

Authors' response:

We appreciate the Reviewer's concerns regarding the heterogeneity of the ligament tissue. 

To clarify, each sample piece was extracted from the center of each full ligament/tendon. For ACL, the sample was taken from the anteromedial bundle. In PCL, however, the bundle structure is not as clearly defined in bovines as in humans so no clear distinction could be made. Since the sample piece was much smaller than the full ligament, there should be very little variation in material and biochemical properties within the sample. Furthermore, since the sample was from mid-substance and not e.g., insertion sites, the chemical and structural properties should be rather homogeneous.

The NIRS measurements were performed on five equispaced locations along the longitudinal axis of the sample piece. These five spectra were then averaged before used to model the relationship between NIR spectrum and the biochemical reference variable. This averaging further helps to eliminate any spectral variation that might exist over the total volume of the sample.

The Reviewer is correct in pointing out that biochemical composition, collagen crimp, and mechanical properties differ between ligament types to some degree. However, overall the tissues are very similar in both general structure and function. By this we mean that each ligament primarily consists of water, hierarchically organized collagen fiber bundles, elastin, and proteoglycans and are loaded in tension. Since the object of this study was to investigate the capability of NIRS to predict compositional (and structural) properties directly and since these properties were explicitly known for each sample, there was little need to separate them by ligament types. For instance, if we want to predict water content in these tissues and know water content for each sample, the information that water content differs between ACL/PCL/LCL/MCL/PT is not necessary. Therefore, we opted to combine all ligament types together. We would like to point out, however, that the ligaments were previously analysed separately by Ristaniemi et al. for biomechanics and biochemistry\\cite{ristaniemi2018comparison, ristaniemi2020comparison}.

Authors' actions:

We now indicate more clearly that the samples originate from the midsection of each ligament and what this means in terms of material properties.

[Page 4, lines 115--119]: A small dumbbell-shaped sample piece (region of interest approximately 2.0 x 1.8 x 10 mm in dimensions) was cut from the midsection of each ligament using a custom punch-tool (Fig. 2 b). Midsection was selected for the sample extraction site as the material properties of the tissue are relatively uniform at this location.

The motivation behind not differentiating between ligament types is now explained in Discussion.

[Page 12, lines 329--337]: Even though the biochemical composition\\cite{ristaniemi2020comparison} and collagen crimp parameters\\cite{ristaniemi2021structure} differ between primary ligaments and PT, ligament type was not included as a predictor in the PLSR models. Rather, each sample was treated as independent ligamentous tissue. This decision was intentional as the main object of the study was to evaluate the capability of NIRS to predict biochemical and structural properties directly. Since all the reference variables were explicitly known for each sample, differentiating them by anatomical location was deemed unecessary. We would like to point out, however, that the ligaments were previously analysed separately by Ristaniemi et al. for biomechanics and biochemistry\\cite{ristaniemi2018comparison, ristaniemi2020comparison}

Reviewer comment 4:

Granted this paper was submitted to a non-clinical journal, however, you frequently mention that NIRS-based evaluation can be used for diagnosis in orthopedic repair surgeries. Can you please elaborate on this? The approach is certainly non-destructive (for the sample) but not non-invasive for patients. For example, clinically if you have a positive Lachman and pivot tests, most surgeons will go ahead and reconstruct the ACL regardless of whether the MRI actually shows a partial or complete rupture, thus, are you hoping to convince clinicians and medical device boards to allow an arthroscopic procedure prior to surgery to identify what state the ligament is in?

Authors' response:

We are delighted in the Reviewers interest towards NIRS-based evaluation in orthopedic repair surgeries.

NIRS-based tissue evaluation for orthopedic repair surgeries has been the overarching goal for most of the research conducted by our group. We do not propose using the technique for a purely diagnostic purposes as that would pose unnecessary risks for the patient. Rather, the general idea is to utilize the technique when a repair procedure (or any kind of arthroscopy) is already being performed. This kind of use case is more intuitive in meniscus repair (or any kind of cartilage injury), where NIRS could be used to quickly and quantitatively locate and map the extent of the injury being repaired. We hypothesize that similar approach could be adopted for ACL reconstruction or rotator cuff surgeries in the future. In the case of a patellar tendon graft, for example, the technique could aid the surgeon in selecting the best location of the donor graft. 

Authors' actions:

A more detailed description of arthroscopic NIRS has now been added to the Introduction.

[Page 3, lines 62--67]: Recently, NIRS has been suggested as a tool for real-time tissue diagnostics in orthopedic applications\\cite{spahn2007near, spahn2010near}. The operating principle of NIRS-based diagnostics is to supplement traditional arthroscopic tissue evaluation methods (i.e., visual observation and manual palpation) with quantitative point measurements made with NIRS. These measurements can be made in real-time and provide more realistic assessment of tissue integrity. Furthermore, the NIRS probe used to perform these measurements can be manufactured in the same shape and size as the traditional arthroscopic hooks.

We now further elaborate as to when the technique should be used in Discussion.

[Page 11, lines 298--303]: Quantitative tissue evaluation during arthroscopy could provide additional diagnostic information during orthopedic repair procedures or help to select optimal rehabilitation procedures after the operation. As this technique is invasive, it should only rarely be performed solely for the diagnosis of injury. Rather, the technique can provide vital supplemental information about the tissue condition when surgical intervention has been decided as treatment.

REFERENCES:

title={Comparison of elastic, viscoelastic and failure tensile material properties of knee ligaments and patellar tendon},

author={Ristaniemi, Aapo and Stenroth, Lauri and Mikkonen, Santtu and Korhonen, Rami K},

journal={Journal of biomechanics},

volume={79},

pages={31--38},

year={2018},

publisher={Elsevier}

}

@article{torniainen2019near,

 title={Near Infrared Spectroscopic Evaluation of Ligament and Tendon Biomechanical Properties},

 author={Torniainen, Jari and Ristaniemi, Aapo and Sarin, Jaakko K and Mikkonen, Santtu and Afara, Isaac O and Stenroth, Lauri and Korhonen, Rami K and T{\"o}yr{\"a}s, Juha},

 journal={Annals of biomedical engineering},

 volume={47},

 number={1},

 pages={213--222},

 year={2019},

 publisher={Springer}

}

@article{ristaniemi2020comparison,

title={Comparison of water, hydroxyproline, uronic acid and elastin contents of bovine knee ligaments and patellar tendon and their relationships with biomechanical properties},

author={Ristaniemi, A and Torniainen, J and Stenroth, L and Finnil{\"a}, MAJ and Paakkonen, T and T{\"o}yr{\"a}s, J and Korhonen, RK},

journal={Journal of the Mechanical Behavior of Biomedical Materials},

pages={103639},

year={2020},

publisher={Elsevier}

}

@article{ristaniemi2021structure,

title={Structure, composition and fibril-reinforced poroviscoelastic properties of bovine knee ligaments and patellar tendon},

author={Ristaniemi, Aapo and Regmi, Dristi and Mondal, Diponkor and Torniainen, Jari and Tanska, Petri and Stenroth, Lauri and Finnil{\"a}, Mikko AJ and T{\"o}yr{\"a}s, Juha and Korhonen, Rami K},

journal={Journal of the Royal Society Interface},

volume={18},

number={174},

pages={20200737},

year={2021},

publisher={The Royal Society}

}

@article{curcio1951near,

title={The near infrared absorption spectrum of liquid water},

author={Curcio, Joseph A and Petty, Charles C},

journal={JOSA},

volume={41},

number={5},

pages={302--304},

year={1951},

publisher={Optical Society of America}

}

@article{sarin2018arthroscopic,

title={Arthroscopic near infrared spectroscopy enables simultaneous quantitative evaluation of articular cartilage and subchondral bone in vivo},

author={Sarin, Jaakko K and Te Moller, Nikae CR and Mancini, Irina AD and Brommer, Harold and Visser, Jetze and Malda, Jos and van Weeren, P Ren{\\'e} and Afara, Isaac O and T{\"o}yr{\"a}s, Juha},

journal={Scientific reports},

volume={8},

number={1},

pages={13409},

year={2018},

publisher={Nature Publishing Group}

}

@article{spahn2007near,

title={Near-infrared (NIR) spectroscopy. A new method for arthroscopic evaluation of low grade degenerated cartilage lesions. Results of a pilot study},

author={Spahn, Gunter and Plettenberg, Holger and Kahl, Enrico and Klinger, Hans M and M{\"u}ckley, Thomas and Hofmann, Gunther O},

journal={BMC musculoskeletal disorders},

volume={8},

number={1},

pages={47},

year={2007},

publisher={BioMed Central}

}

@article{spahn2010near,

title={Near-infrared spectroscopy for arthroscopic evaluation of cartilage lesions: results of a blinded, prospective, interobserver study},

author={Spahn, Gunter and Klinger, Hans Michael and Baums, Mike and Hoffmann, Martin and Plettenberg, Holger and Kroker, Anne and Hofmann, Gunther O},

journal={The American journal of sports medicine},

volume={38},

number={12},

pages={2516--2521},

year={2010},

publisher={Sage Publications Sage CA: Los Angeles, CA}

}

@article{afara2021characterization,

title={Characterization of connective tissues using near-infrared spectroscopy and imaging},

author={Afara, Isaac O and Shaikh, Rubina and Nippolainen, Ervin and Querido, William and Torniainen, Jari and Sarin, Jaakko K and Kandel, Shital and Pleshko, Nancy and T{\"o}yr{\"a}s, Juha},

journal={Nature Protocols},

volume={16},

number={2},

pages={1297--1329},

year={2021},

publisher={Nature Publishing Group}

}

@article{ristaniemi2021biomechanical,

title={Biomechanical, biochemical, and near infrared spectral data of bovine knee ligaments and patellar tendon},

author={Ristaniemi, Aapo and Torniainen, Jari and Paakkonen, Tommi and Stenroth, Lauri and Finnil{\"a}, Mikko AJ and Tanska, Petri and T{\"o}yr{\"a}s, Juha and Korhonen, Rami K},

journal={Data in Brief},

volume={36},

pages={106976},

year={2021},

publisher={Elsevier}

}

@article{apostolakos2014enthesis,

title={The enthesis: a review of the tendon-to-bone insertion},

author={Apostolakos, John and Durant, Thomas JS and Dwyer, Corey R and Russell, Ryan P and Weinreb, Jeffrey H and Alaee, Farhang and Beitzel, Knut and McCarthy, Mary Beth and Cote, Mark P and Mazzocca, Augustus D},

journal={Muscles, ligaments and tendons journal},

volume={4},

number={3},

pages={333},

year={2014},

publisher={CIC Edizioni internazionali}

}

---

## [Decision Letter · Decision Letter 1]

17 Jan 2022

Near Infrared Spectroscopic Evaluation of Biochemical and Crimp Properties of Knee Joint Ligaments and Patellar Tendon

PONE-D-21-18102R1

Dear Dr. Torniainen,

We’re pleased to inform you that your manuscript has been judged scientifically suitable for publication and will be formally accepted for publication once it meets all outstanding technical requirements.

Kind regards,

Manabu Sakakibara, Ph.D.

Academic Editor

PLOS ONE

Additional Editor Comments (optional):

Reviewers' comments:

Reviewer's Responses to Questions

**Comments to the Author**

1. If the authors have adequately addressed your comments raised in a previous round of review and you feel that this manuscript is now acceptable for publication, you may indicate that here to bypass the “Comments to the Author” section, enter your conflict of interest statement in the “Confidential to Editor” section, and submit your "Accept" recommendation.

Reviewer #1: All comments have been addressed

Reviewer #2: All comments have been addressed

2. Is the manuscript technically sound, and do the data support the conclusions?

Reviewer #1: (No Response)

Reviewer #2: Yes

3. Has the statistical analysis been performed appropriately and rigorously? 

Reviewer #1: (No Response)

Reviewer #2: Yes

4. Have the authors made all data underlying the findings in their manuscript fully available?

Reviewer #1: (No Response)

Reviewer #2: Yes

5. Is the manuscript presented in an intelligible fashion and written in standard English?

Reviewer #1: (No Response)

Reviewer #2: Yes

6. Review Comments to the Author

Reviewer #1: (No Response)

Reviewer #2: The authors have addressed all the queries and therefore I support publication without further changes.

7. PLOS authors have the option to publish the peer review history of their article (what does this mean?). If published, this will include your full peer review and any attached files.

Reviewer #1: No

Reviewer #2: No

---

## [Editor Report · Acceptance letter]

4 Feb 2022

PONE-D-21-18102R1 

Near infrared spectroscopic evaluation of biochemical and crimp properties of knee joint ligaments and patellar tendon 

Dear Dr. Torniainen:

I'm pleased to inform you that your manuscript has been deemed suitable for publication in PLOS ONE. Congratulations! Your manuscript is now with our production department. 

Kind regards, 

on behalf of

Dr. Manabu Sakakibara 

Academic Editor

PLOS ONE